# Comparative Effectiveness of Treatments for Shoulder Subluxation After Stroke: A Systematic Review and Network Meta-Analysis

**DOI:** 10.3390/jcm14196913

**Published:** 2025-09-29

**Authors:** Jong-Mi Park, Hee-Jae Park, Seo-Yeon Yoon, Yong-Wook Kim, Jae-Il Shin, Sang-Chul Lee

**Affiliations:** 1Department and Research Institute of Rehabilitation Medicine, Yonsei University College of Medicine, Seoul 03722, Republic of Korea; pjm0207@yuhs.ac (J.-M.P.); hjpark9401@gmail.com (H.-J.P.);; 2Department of Pediatrics, Yonsei University College of Medicine, Seoul 03722, Republic of Korea; 3The Center for Medical Education Training and Professional Development in Yonsei-Donggok Medical Education Institute, Seoul 03722, Republic of Korea; 4Severance Underwood Meta-Research Center, Institute of Convergence Science, Yonsei University, Seoul 03722, Republic of Korea

**Keywords:** stroke, shoulder dislocation, electric stimulation, orthotic devices, network meta-analysis

## Abstract

**Background**: Shoulder subluxation and pain are common complications of stroke that impair upper limb function. **Objectives**: This study conducted a systematic review and network meta-analysis to compare multiple therapeutic interventions for post-stroke shoulder subluxation, establishing an evidence-based hierarchy of treatment efficacy to optimize rehabilitation strategies and guide clinical practice. **Methods**: A comprehensive search was conducted using the MEDLINE, EMBASE, Cochrane, Scopus, and Web of Science databases until 8 August 2025. Randomized controlled trials evaluating treatments for shoulder subluxation, including neuromuscular electrical stimulation (NMES), Kinesio taping, corticosteroid injections, slings, repetitive peripheral magnetic stimulation, and electroacupuncture, were included. The follow-up duration in the included trials ranged from 1 to 12 weeks. Effect sizes were calculated using standardized mean differences with a random-effects model, and treatment rankings were determined using surface under the cumulative ranking curve (SUCRA). **Results**: Thirteen studies including 402 patients were analyzed. NMES was the most effective intervention for reducing subluxation distance (SUCRA: 84.9), while corticosteroid injections provided the greatest pain relief at rest (SUCRA: 73.6). Kinesio taping was most effective for functional recovery, as measured by the Fugl–Meyer Assessment (SUCRA: 98.5), and for pain relief during activity (SUCRA: 87.7). **Conclusions**: Our network meta-analysis suggests that different interventions are optimal for specific aspects of post-stroke shoulder impairment. NMES most effectively reduces subluxation distance, whereas corticosteroid injections are most effective for alleviating pain at rest. Kinesio taping appears superior for enhancing functional recovery and reducing pain during movement. These findings, based on short-term follow-up durations (1–12 weeks), provide an evidence-based ranking of interventions to support multimodal rehabilitation and inform clinical decision-making. The observed heterogeneity across studies underscores the need for standardized treatment protocols and rigorous long-term investigations.

## 1. Introduction

Shoulder subluxation is a common complication that can occur in 17–81% of patients with hemiplegia following a stroke [1]. Clinically, it can be identified by palpating the space between the acromion and the head of the humerus, where an increase in distance indicates subluxation [2]. This condition arises due to the weakening of the shoulder muscles, leading to a downward displacement of the humeral head from the glenoid fossa under the influence of gravity [3].

Shoulder subluxation can contribute to limited range of motion, potentially slowing upper limb functional recovery. In addition, it can cause complex regional pain syndrome, soft tissue injury, and pain [4,5]. Over time, patients may experience increased spasticity, leading to further movement limitations and pain [4]. This condition can extend the duration of hospital stay and negatively impact the patient’s psychological well-being [6]. If left untreated, shoulder subluxation may be irreversible [7]. Moreover, shoulder pain can also significantly hinder functional recovery, contribute to depression, and reduce the overall quality of life [8].

Therefore, it is crucial to manage shoulder subluxation and pain promptly. Currently, various methods are known for managing shoulder subluxation, including neuromuscular electrical stimulation (NMES), supporting devices, strapping, electroacupuncture, and repetitive Peripheral Magnetic Stimulation (rPMS). For pain management, a wide range of techniques is available, such as physical therapy, taping, anesthetic suprascapular nerve block, intramuscular injections of botulinum toxin type A, corticosteroid injections, segmental neuromyotherapy, trigger-point dry needling, robotic-assisted shoulder rehabilitation therapy, platelet-rich plasma injection, repetitive transcranial magnetic stimulation, peripheral nerve stimulation, transcutaneous electrical nerve stimulation, NMES, and interferential current stimulation [9,10,11,12].

This study seeks to comprehensively evaluate the effectiveness of diverse interventions for post-stroke shoulder subluxation. While previous systematic reviews, such as that by Arya et al. (2018) [9], have summarized the available evidence, most randomized controlled trials (RCTs) have compared only a limited number of treatments, leaving uncertainty regarding their relative efficacy across the full spectrum of available options. To address this gap, our study applies a network meta-analysis (NMA) to simultaneously compare multiple therapeutic approaches. By integrating both direct and indirect evidence, this NMA provides an evidence-based hierarchy of treatment efficacy, offering insights that may guide clinical decision-making, optimize individualized rehabilitation strategies, and inform the development of future practice guidelines.

## 2. Materials and Methods

This study followed the Preferred Reporting Items for Systematic Reviews and Meta-Analyses (PRISMA) 2020 guidelines for its conduct and reporting (Appendix A) [13,14]. This review was registered in the International Prospective Register of Systematic Reviews (PROSPERO) on 14 February 2025 (registration number: CRD42025560231). Institutional review board approval was not required for this study.

### 2.1. Search Strategy

We conducted a systematic search across multiple databases, including Medline, EMBASE, Cochrane, Scopus, and Web of Science, covering all records from inception until 5 July 2024. An updated search was performed on 8 August 2025. To identify additional relevant studies, the reference lists of the included studies and related review articles were reviewed. The detailed search strategy is provided in Appendix A.

### 2.2. Inclusion and Exclusion Criteria

This study applied the PICOS (Population, Intervention, Comparison, Outcomes, and Study Design) framework to define the inclusion criteria. The study population comprised patients diagnosed with shoulder subluxation after stroke. The interventions included various treatment approaches for shoulder subluxation, such as slings, Kinesio taping, NMES, rPMS, local steroid injections, and electroacupuncture. The comparison group consisted of patients receiving conventional treatment or a sham intervention. Conventional treatment was defined as a standard rehabilitation program, including physical and occupational therapy, that did not involve the specific active interventions being compared in the trials. Conventional rehabilitation comprised passive and active range of motion exercises, stretching, positioning, strengthening, and task-oriented training, with some protocols also including posture management, balance or gait training, and complication prevention. Programs were typically administered 5 days per week for 30–180 min per day over 4–6 weeks. Studies that directly compared different active interventions were also included.

The primary outcomes were shoulder subluxation distance, measured radiographically or ultrasonographically using indices such as the Acromio-Humeral Distance (AHD), Acromio-Humeral Interval (AHI), Acromion–Greater Tuberosity (AGT) distance, or Acromion–Lesser Tuberosity (ALT) distance, as well as by clinical palpation methods (e.g., fingerbreadth technique); pain, assessed with the Visual Analog Scale (VAS), Numerical Rating Scale (NRS), or Numerical Pain Rating Scale (NPRS); and functional recovery, primarily evaluated using the Fugl–Meyer Assessment (FMA), but also reported in some studies using additional scales such as the Shoulder Pain and Disability Index (SPADI) and the Modified Barthel Index (MBI). Another primary outcome was shoulder passive range of motion (PROM), including flexion, abduction, and external rotation; however, due to insufficient data for flexion and external rotation, the final analysis was restricted to abduction. The secondary outcomes included adverse events associated with the interventions. For the study design, only RCTs were included.

### 2.3. Data Extraction and Quality Assessment

Two reviewers independently extracted relevant data from the included studies using a predefined data extraction table. Any discrepancies were resolved through discussion with a third reviewer. The extracted data included author name, year of publication, study design, sample size, population characteristics (mean age, sex, time since stroke, and baseline functional scores), intervention and control details, and primary and secondary outcomes. If the reported data were unclear or incomplete, the authors were contacted to obtain additional unpublished information. The Revised Cochrane Risk of Bias Tool for Randomized Trials (RoB2) was used to assess the risk of bias [15]. We assessed the certainty of evidence for the main outcomes using Confidence in Network Meta-Analysis (CINeMA) [16].

### 2.4. Data Synthesis and Statistical Analysis

The mean and standard deviation (SD) of the change from baseline for both the treatment and control groups were calculated as described in Chapter 6 of the Cochrane Handbook (version 6.4) [17] when necessary for studies with insufficient data. Effect sizes were expressed as standardized mean differences (SMDs) with 95% confidence intervals (CIs) using a random effects model. Pairwise meta-analyses were conducted for comparisons with at least two trials, and statistical heterogeneity was assessed using I^2^ statistics, where I^2^ ≥ 50% indicated substantial heterogeneity. A random-effects NMA was performed when no major violations of transitivity were detected. The consistency of the network was examined by comparing direct and indirect treatment effects using node-splitting techniques and the design-by-treatment interaction model [18,19]. The ranking of treatment efficacy was determined using the surface under the cumulative ranking curve (SUCRA). However, SUCRA values should not be interpreted in isolation, as they provide only a relative hierarchy of interventions. They must be considered alongside the magnitude of the SMDs and the precision of its 95% CIs to determine the clinical significance and certainty of the findings, particularly when confidence intervals are wide [20]. Publication bias was assessed using funnel plot symmetry and Egger’s test. All statistical analyses, including NMA, were performed using the network and network graphs packages in the STATA software (version 18.0; StataCorp, College Station, TX, USA) by applying a frequentist approach [19,21].

## 3. Results

### 3.1. Study Identification and Characteristics

A flowchart illustrating the study selection process is shown in Figure 1. Of the 483 initially screened studies, 106 duplicates were excluded. After reviewing the titles and abstracts, 218 and 81 studies were excluded, respectively. Among the 78 remaining studies, 65 were excluded for various reasons, including inappropriate population (*n* = 14), non-English language (*n* = 4), lack of full text (*n* = 23), unsuitable outcome measurements (*n* = 3), inappropriate interventions (*n* = 18), study protocols (*n* = 2), and full-text could not be obtained (*n* = 1). Ultimately, 13 studies met the eligibility criteria and were included in the meta-analysis, comprising 402 patients, with sample sizes ranging from 9 [22] to 25 [23] participants per study. The average age of the participants ranged from 51 [24] to 73.3 [25] years, and all studies included both male and female participants. The follow-up duration varied from one to 12 weeks. The included studies spanned acute [26] to chronic post-stroke stages [27], with baseline FMA-UE scores generally indicating severe impairment (ranging from 7.13 [22] to 30.2 [12] out of 66). The included interventions consisted of two studies on slings [22,25], three on Kinesio taping [27,28,29], six on NMES [23,25,26,30,31,32], one on local steroid injections [33], one on rPMS [12], and one on electroacupuncture [24]. The intervention parameters for the NMES and rPMS trials are summarized in Table 1. Protocols varied considerably, particularly in session duration and frequency, with daily treatment times ranging from 20 min to several hours. Full details remain available in Appendix A.

### 3.2. Risk of Bias and Publication Bias Assessment

All 13 studies were randomized trials that provided details of their randomization methods. Only three studies [24,27,29] implemented a sham intervention in the control group, thus minimizing bias related to the intended intervention. Missing outcome data bias was reported in four studies [12,25,31,33], whereas six studies were identified as having a risk of bias in their outcome measurements [22,23,30,31,32,33]. Additionally, only six studies had a pre-registered protocol, indicating a low risk of bias in the selection of reported results [12,22,24,25,27,29]. Ultimately, the analysis included three RCTs with a low risk of bias [24,27,29], eight RCTs with some concerns [12,22,23,25,26,28,30,32], and two RCTs with a high risk of bias [31,33]. Appendix A presents a traffic light diagram summarizing the bias assessment for each included study. Funnel plots (Appendix A) and Egger’s test *p*-values indicated significant asymmetry for the shoulder subluxation distance (*p* = 0.006). However, trim-and-fill analysis did not impute any missing studies, and the pooled effect estimate remained unchanged, suggesting that the observed asymmetry is unlikely to be fully explained by publication bias alone and may instead reflect heterogeneity among the included trials.

### 3.3. Evaluation of Inconsistency

Inconsistency was evaluated using both local (node-splitting) and global (de-sign-by-treatment interaction) approaches. No significant local inconsistency was observed for any treatment comparison (Appendix A); however, the global analysis identified a significant inconsistency for PROM Shoulder Abduction (χ^2^ = 8.18, *p* = 0.004) (Appendix A), indicating violation of the transitivity assumption; thus, the pooled estimates and SUCRA rankings for this outcome should be regarded as unreliable and exploratory only.

### 3.4. Effects of Shoulder Subluxation Distance

A network plot and league table illustrate the comparative effects of interventions on shoulder subluxation distance (Figure 2a, Table 2A). The analysis of treatment efficacy, based on SUCRA values, ranked NMES as the most effective intervention (SUCRA: 84.9), followed by sling and rPMS (both 57.9) (Figure 3a, Appendix A). When compared with standard treatment, only NMES demonstrated a statistically significant reduction in shoulder subluxation distance (SMD: −2.21; 95% Confidence Interval, CI: −4.34 to −0.08). Other interventions did not show a statistically significant advantage over standard treatment for this outcome Figure 4a.

### 3.5. Effects of Shoulder Pain

A network plot and corresponding league tables illustrating the effects of shoulder pain at rest and during activity are presented in Figure 2b,c, and Table 2B,C, respectively. For pain at rest, intra-articular steroid injection was the most effective treatment (SUCRA: 73.6), followed by electroacupuncture (58.1) (Figure 3b, Appendix A). However, when compared to standard treatment, no intervention achieved a statistically significant reduction in pain at rest Figure 4b. For pain during activity, Kinesio taping ranked highest in efficacy (SUCRA: 87.7), followed by steroid injection (82.5) (Figure 3c, Appendix A). Both taping (SMD: −1.04; 95% CI: −2.00 to −0.08) and steroid injection (SMD: −0.81; 95% CI: −1.28 to −0.34) were significantly more effective than standard treatment in reducing pain during activity Figure 4c.

### 3.6. Effects of Functional Recovery Measured by FMA

A network plot and corresponding league table for FMA are presented in Figure 2d and Table 2D. For improving function as measured by the FMA, Kinesio taping was found to be the most effective treatment with a SUCRA value of 98.5, followed by NMES (55.4) (Figure 3d, Appendix A). In direct comparison to other interventions, taping was significantly more effective than both the sling (SMD: 1.75; 95% CI: 0.70 to 2.79) and NMES (SMD: 1.08; 95% CI: 0.31 to 1.85) Table 2D. Conversely, the sling was significantly less effective than standard treatment (SMD: −0.78; 95% CI: −1.36 to −0.20) Figure 4d.

### 3.7. Effects of Passive Range of Motion of the Shoulder

The PROM results are presented for transparency but must be regarded as exploratory only, as significant global inconsistency (χ^2^ = 8.18, *p* = 0.004) was detected in this network. Therefore, these findings should not be interpreted as definitive evidence of comparative efficacy. Initially, we aimed to analyze the PROM of the shoulder—including flexion, abduction, and external rotation—as an outcome. However, due to the limited number of studies re-porting these measures, we confined our analysis to the PROM of shoulder abduction. A network plot and league tables for shoulder abduction PROM is shown in Figure 2e and Table 2E. Steroid intra-articular injection was the most effective intervention for this out-come (SUCRA: 95.1), followed by the sling (57.3) (Figure 3e, Appendix A). Steroid injections showed a statistically significant improvement in shoulder abduction PROM when com-pared to standard treatment (SMD: 2.48; 95% CI: 1.68 to 3.28), slings (SMD: 1.80; 95% CI: 1.15 to 2.46), and NMES (SMD: 1.91; 95% CI: 0.77 to 3.04) Table 2E. Slings also demonstrated a significant improvement over standard treatment (SMD: 0.68; 95% CI: 0.21 to 1.15) Figure 4e.

### 3.8. Safety/Adverse Events

Across the included studies, no serious adverse events were reported, and most interventions, including NMES, rPMS, Kinesio taping, electroacupuncture, and steroid injections, were well tolerated. While some studies reported mild discomfort during electrical stimulation, particularly due to electrode placement or muscle contractions [12,26,30], no significant complications, such as skin irritation, infections, or tissue damage, were documented. However, as some studies lacked explicit adverse event reporting, future research should ensure systematic documentation to provide a more comprehensive safety profile.

### 3.9. Grading of Evidence

According to the CINeMA assessment (Appendix A), the certainty of evidence was generally moderate for shoulder subluxation distance and pain during activity, with downgrades mainly due to imprecision. Confidence for pain at rest and functional outcomes (FMA) was low, reflecting within-study bias (as captured by RoB2) and concerns about selective reporting. For PROM shoulder abduction, ratings were low to very low because of both significant network inconsistency and sparse data, limiting the reliability of these estimates.

## 4. Discussion

This NMA assessed the effectiveness of various treatments for shoulder subluxation after stroke. NMES demonstrated the highest efficacy in reducing the subluxation distance (SUCRA: 84.9), whereas intra-articular steroid injection was most effective for alleviating pain at rest (SUCRA: 73.6). Kinesio taping showed the greatest benefit for pain during activity (SUCRA: 87.7). Additionally, Kinesio taping ranked highest for functional recovery, as measured by the FMA (SUCRA: 98.5), indicating its role in improving motor function. Steroid injection was the most effective intervention for shoulder abduction PROM (SUCRA: 95.1). These findings suggest that NMES, Kinesio taping, and steroid injections are among the most effective interventions for managing the different aspects of shoulder subluxation and associated impairments in patients with stroke.

Our findings are broadly consistent with, and extend, those of prior systematic reviews and meta-analyses. In line with Lee et al. (2017), our NMA identified NMES as the most effective intervention for reducing shoulder subluxation in acute and subacute stroke patients [5]. However, whereas our analysis did not demonstrate a significant benefit of NMES for pain relief, Wang et al. (2025) reported that functional electrical stimulation (FES), a subtype of NMES, was effective in alleviating hemiplegic shoulder pain [34]. This divergence may reflect differences in stimulation protocols, outcome measures, or patient populations, underscoring the need for further investigation into optimal NMES/FES dosing strategies. With respect to Kinesio taping, our NMA found it to be the most effective intervention for enhancing functional recovery and reducing pain during activity. These results partially align with the systematic review by Ravichandran et al. (2019), which concluded that taping reduces both pain and shoulder subluxation [35]. While both reviews support the role of taping in pain management, our analysis suggests that its primary benefits may lie in functional improvements and activity-related pain relief rather than structural correction of subluxation. By employing a NMA, our study synthesizes these findings within a single comparative framework and provides the first quantitative, hierarchical ranking of multiple interventions. This approach offers a more comprehensive guide for clinicians seeking to optimize rehabilitation strategies for post-stroke shoulder impairment.

The strong performance of NMES in reducing subluxation distance is likely attributable to its mechanism of action. NMES activates key stabilizing muscles such as the supraspinatus and posterior deltoid to maintain glenohumeral alignment, enhances muscle strength and endurance, prevents capsular stretching and elongation, and promotes sensorimotor integration and cortical reorganization [5,23,30]. Early application (within 3 months post-stroke) is crucial for preventing progressive subluxation [26]. Despite the clear benefits of NMES in reducing subluxation, it did not have significant effects on arm function or shoulder pain relief, highlighting the need for multimodal rehabilitation approaches to address broader functional impairments.

Kinesio taping also shows promise as an adjunctive intervention. When combined with conventional rehabilitation, taping can reduce pain, improve shoulder range of motion, and enhance overall function [36]. The mechanism behind its effectiveness is multifaceted: it provides proprioceptive feedback, supports joint stability, and modulates pain perception. The elasticity of the tape allows for continuous skin stimulation, which may improve circulation, reduce pressure on pain receptors, enhance neuromuscular activation, and promote lymphatic drainage [37]. These effects contribute to pain relief during activity and functional improvement. Additionally, Kinesio taping applied to the back muscles during trunk rehabilitation increases forward mobility in patients with stroke, potentially improving postural control and sitting stability [38].

Corticosteroid injections have been widely used to alleviate hemiplegic shoulder pain and improve range of motion in patients with stroke. Multiple studies have reported significant reductions in pain and improvements in shoulder flexion, abduction, and external rotation [33,39,40]. Various injection techniques, including intra-articular steroid injection, suprascapular nerve block, and subacromial corticosteroid injection, have been found to be effective, although no single method has proven superior [41,42]. The anti-inflammatory properties of corticosteroids help reduce joint inflammation, capsular stiffness, and impingement-related pain, leading to pain relief that can last for up to 8 weeks post-treatment [39].

Interestingly, although shoulder pain is often assumed to be related to subluxation, research suggests that pain is more closely associated with restricted range of motion and soft tissue involvement than with the severity of subluxation [33,43]. Our study aligns with these findings, showing that NMES was effective in reducing subluxation, whereas corticosteroid injections primarily improved pain and range of motion without significantly affecting subluxation. These results highlight the complex and multifactorial relationships between shoulder pain, subluxation, and functional recovery in patients with hemiplegia. Although steroid injections serve as an effective short-term intervention, their limited impact on functional outcomes and potential risks, such as tendon degeneration and joint instability, emphasize the need for their judicious use within a comprehensive rehabilitation program, incorporating physical therapy and neuromuscular training for long-term recovery [33,39].

The included trials covered a wide range of stroke onset durations, from the acute phase (<1 month [26,30]) to the chronic phase (>12 months [27,33]). This heterogeneity likely contributed to differences in treatment responsiveness. In the acute and early subacute stages, NMES was consistently effective in preventing or reducing subluxation distance and improving motor activation [10,26,30]. These findings support the notion that early stimulation may help maintain glenohumeral alignment before irreversible capsular stretching and soft-tissue changes occur. In contrast, interventions tested predominantly in later phases showed a shift in therapeutic goals. Taping studies conducted in chronic populations [27] reported improvements in pain, active range of motion, and functional outcomes but limited structural correction of subluxation. Similarly, steroid injection trials were conducted largely in chronic or late-subacute patients [33], with benefits focused on pain reduction and range of motion rather than alignment. These patterns suggest a timing-by-treatment interaction: NMES may be most beneficial when introduced early to prevent progressive subluxation, whereas taping and steroid injections appear more suited to managing established pain and functional limitations in chronic stages. Although our dataset did not allow a statistically reliable subgroup network analysis, the stratified narrative underscores stroke phase as a plausible effect modifier and highlights the need for future RCTs designed to directly test timing-specific treatment efficacy.

This NMA has several strengths. First, it provides a comprehensive overview of the various treatments for shoulder subluxation after stroke, allowing for both direct and indirect comparisons between interventions. Our study included a wide range of treatments, from conventional therapies to more recent interventions, such as repetitive peripheral magnetic stimulation, offering a broad perspective on available options. Additionally, we employed a rigorous methodology, including a thorough literature search, careful study selection, and robust statistical analyses, to enhance the reliability of our findings.

Several limitations of this review should be carefully considered. First, heterogeneity in patient characteristics, intervention protocols, and outcome measures across the included studies may have influenced the results. In particular, intra-articular and subacromial steroid injections were pooled as a single category (“steroid”), which may have introduced clinical heterogeneity because of differences in injection targets and techniques. Given the limited number of available trials, subgroup or sensitivity analyses were not feasible, and our findings regarding steroid efficacy should therefore be interpreted with caution. Furthermore, the possibility of publication bias cannot be fully excluded. Specifically, for the shoulder subluxation outcome, a significant Egger’s test suggested the presence of small-study effects. While a subsequent trim-and-fill analysis did not alter the pooled estimate—implying the asymmetry is likely driven by heterogeneity rather than missing studies—the potential for bias from small, influential trials remains. Therefore, the findings regarding NMES efficacy for subluxation must be interpreted with caution.

More importantly, all included trials had short follow-up durations (1–12 weeks), restricting our conclusions to immediate or short-term effects and leaving the long-term sustainability of these interventions uncertain. As noted in the Methods, the analysis was constrained by limited data on ROM. The NMA could only be performed for shoulder abduction, as data for flexion and external rotation—movements critical for daily activities and pain mechanisms [44,45]—were rarely reported in the included trials. In addition, the minimal clinically important difference (MCID) for shoulder ROM in post-stroke patients has not been established, making it difficult to determine whether the statistically significant improvements observed translate into clinically meaningful benefits. Future studies should therefore employ standardized and comprehensive ROM assessments, establish clinically relevant thresholds for functional recovery, and investigate long-term outcomes.

## 5. Conclusions

This network meta-analysis provides an evidence-based hierarchy for managing post-stroke shoulder impairment, indicating that NMES is most effective for reducing subluxation, Kinesio taping for enhancing function and alleviating activity-related pain, and steroid injections for relieving pain at rest. However, these findings should be interpreted with caution, as the evidence for several interventions is derived from single studies, and no firm conclusions can be drawn regarding range of motion outcomes due to inconsistent data. Overall, our results support a tailored, multimodal rehabilitation approach that aligns therapeutic choices with specific patient goals while acknowledging the limitations of the current evidence base.

## Figures and Tables

**Figure 1 jcm-14-06913-f001:**
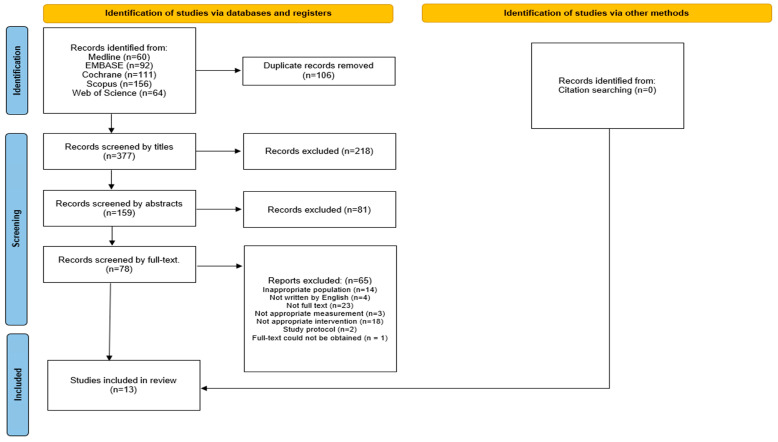
A PRISMA (Preferred Reporting Items for Systematic Reviews and Meta-Analyses) flowchart illustrating the screening and selection process of the included studies.

**Figure 2 jcm-14-06913-f002:**
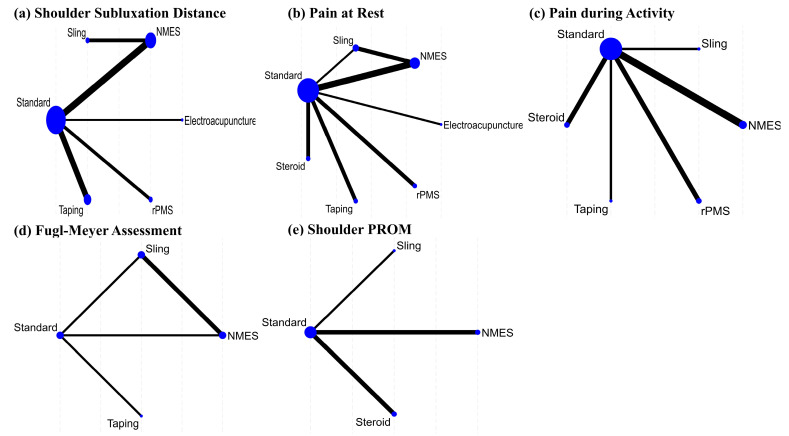
Network plots of treatments for shoulder subluxation following stroke on (**a**) Shoulder subluxation distance, (**b**) Pain at rest, (**c**) Pain during activity, (**d**) Fugl–Meyer assessment, and (**e**) shoulder PROM.

**Figure 3 jcm-14-06913-f003:**
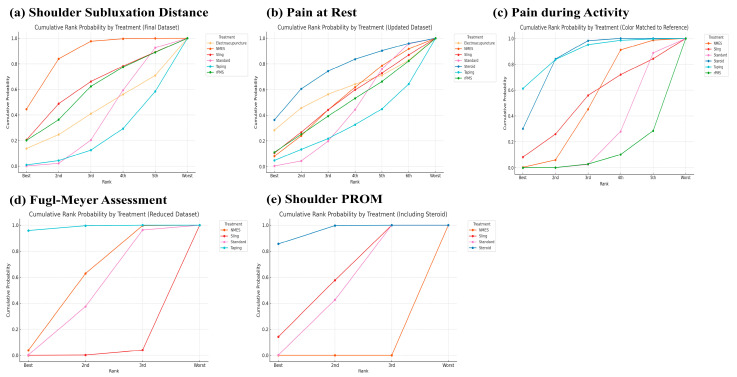
Cumulative rank probability by each treatment on (**a**) Shoulder subluxation distance, (**b**) Pain at rest, (**c**) Pain during activity, (**d**) Fugl–Meyer assessment, and (**e**) shoulder PROM.

**Figure 4 jcm-14-06913-f004:**
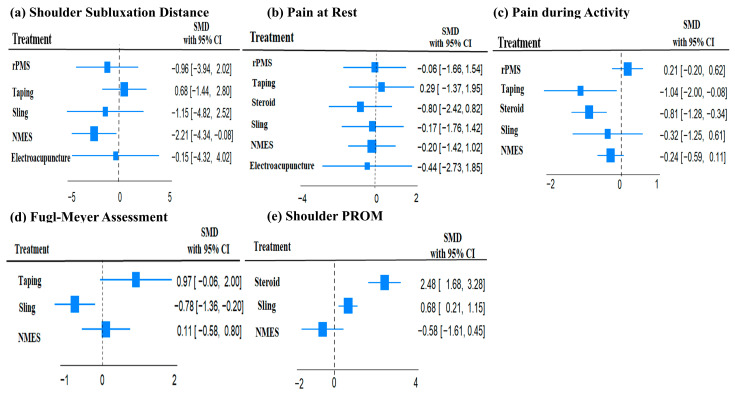
Forest plots of the standard mean difference and 95% confidence interval of each treatment compared with standard treatment on (**a**) Shoulder subluxation distance, (**b**) Pain at rest, (**c**) Pain during activity, (**d**) Fugl–Meyer assessment, and (**e**) shoulder PROM.

**Table 1 jcm-14-06913-t001:** Intervention parameters for NMES and rPMS in the included trials.

Author (Year)	Intervention	Frequency (Hz)	Pulse Width (µs)	Session Duration & Frequency	Target Muscles
Canan Turkkan (2017) [32]	NMES	25	250	60 min/session, 5 days/wk, 4 wks (total 20 sessions)	Supraspinatus, upper trapezius, posterior deltoid
Chen Lavi (2022) [25]	NMES	35	250	30 min first week, gradually increasing by 10 min each week up to a max of 60 min/period from week 4), 5 days/wk, 6 wks (up to 3 h/day)	Supraspinatus, posterior deltoid
Engin Koyuncu (2010) [23]	FES	36	250	1 h/day, 5 times/wk, 4 wks (total 20 sessions)	Supraspinatus, posterior deltoid
Pouran D. Faghri (1994) [30]	FES	35	NR	1.5 to 6 h/day, 7 days/wk, 6 wks. Muscle contraction/relaxation ratio gradually increased (from 10/12 s ON-OFF to 30/2 s ON-OFF)	Posterior deltoid (active electrode), Supraspinatus (passive electrode)
Sandra L. Linn (1999) [26]	ES	30	300	30–60 min/session, 4 sessions/day, 4 wks (30 min week 1, 45 min weeks 2–3, 60 min week 4)	Supraspinatus, posterior deltoid
Ozgur Z. Karaahmet (2018) [31]	FES-cycling	20	300	30 min/day (5 min warm-up, 15 min FES-cycling, 5 min cool-down), 5 times/wk, 4 wks (total 20 sessions)	Anterior deltoid, posterior deltoid, biceps, triceps
Kenta Fujimura (2024) [12]	rPMS	30	350	20 min/day (2 s ON @30 Hz, 3 s OFF, total 6000 pulses/muscle), 6 wks	Supraspinatus, posterior deltoid/infraspinatus

NMES, neuromuscular electrical stimulation; wk, week; rPMS, repetitive peripheral magnetic stimulation; FES, functional electrical stimulation; ES, electrical stimulation; NR, not reported.

**Table 2 jcm-14-06913-t002:** League table for all standard mean differences and 95% confidence interval by outcome and intervention.

(A) Shoulder Subluxation Distance						
rPMS	1.64 (−2.01,5.29)	No data	−0.18 (−4.91,4.55)	−1.24 (−4.91,2.42)	0.81 (−4.31,5.93)	0.96 (−2.02,3.94)
−1.64 (−5.29,2.01)	Taping	No data	−1.83 (−6.07,2.42)	−2.89 (−5.89,0.12)	−0.83 (−5.50,3.84)	−0.68 (−2.80,1.44)
No data	No data	Steroid	No data	No data	No data	No data
0.18 (−4.55,4.91)	1.83 (−2.42,6.07)	No data	Sling	1.06 (−1.93,4.05)	1.00 (−4.56,6.55)	1.15 (−2.53,4.82)
1.24 (−2.42,4.91)	2.89 (−0.12,5.89)	No data	−1.06 (−4.05,1.93)	NMES	2.06 (−2.62,6.74)	**2.21 (0.08,4.34) ***
−0.81 (−5.93,4.31)	0.83 (−3.84,5.50)	No data	−1.00 (−6.55,4.56)	−2.06 (−6.74,2.62)	Electroacupuncture	0.15 (−4.02,4.32)
−0.96 (−3.94,2.02)	0.68 (−1.44,2.80)	No data	−1.15 (−4.82,2.53)	**−2.21 (−4.34,−0.08) ***	−0.15 (−4.32,4.02)	Standard
**(B) Pain at Rest**						
rPMS	0.35 (−1.96,2.65)	−0.74 (−3.02,1.53)	−0.11 (−2.37,2.14)	−0.14 (−2.15,1.87)	−0.39 (−3.18,2.41)	0.06 (−1.54,1.66)
−0.35 (−2.65,1.96)	Taping	−1.09 (−3.41,1.23)	−0.46 (−2.76,1.84)	−0.49 (−2.55,1.57)	−0.73 (−3.57,2.10)	−0.29 (−1.95,1.38)
0.74 (−1.53,3.02)	1.09 (−1.23,3.41)	Steroid	0.63 (−1.64,2.89)	0.60 (−1.42,2.62)	0.36 (−2.45,3.16)	0.80 (−0.82,2.42)
0.11 (−2.14,2.37)	0.46 (−1.84,2.76)	−0.63 (−2.89,1.64)	Sling	−0.03 (−1.45,1.39)	−0.27 (−3.06,2.52)	0.17 (−1.42,1.76)
0.14 (−1.87,2.15)	0.49 (−1.57,2.55)	−0.60 (−2.62,1.42)	0.03 (−1.39,1.45)	NMES	−0.24 (−2.84,2.35)	0.20 (−1.02,1.42)
0.39 (−2.41,3.18)	0.73 (−2.10,3.57)	−0.36 (−3.16,2.45)	0.27 (−2.52,3.06)	0.24 (−2.35,2.84)	Electroacupuncture	0.44 (−1.85,2.74)
−0.06 (−1.66,1.54)	0.29 (−1.38,1.95)	−0.80 (−2.42,0.82)	−0.17 (−1.76,1.42)	−0.20 (−1.42,1.02)	−0.44 (−2.74,1.85)	Standard
**(C) Pain During Activity**						
rPMS	**−1.25 (−2.30,−0.20) ***	**−1.01 (−1.64,−0.39) ***	−0.52 (−1.54,0.49)	−0.45 (−0.98,0.09)	No data	−0.21 (−0.62,0.20)
**1.25 (0.20,2.30) ***	Taping	0.24 (−0.84,1.31)	0.73 (−0.61,2.07)	0.80 (−0.22,1.83)	No data	**1.04 (0.08,2.00) ***
**1.01 (0.39,1.64) ***	−0.24 (−1.31,0.84)	Steroid	0.49 (−0.55,1.53)	0.57 (−0.02,1.15)	No data	**0.81 (0.34,1.28) ***
0.52 (−0.49,1.54)	−0.73 (−2.07,0.61)	−0.49 (−1.53,0.55)	Sling	0.08 (−0.92,1.07)	No data	0.32 (−0.61,1.25)
0.45 (−0.09,0.98)	−0.80 (−1.83,0.22)	−0.57 (−1.15,0.02)	−0.08 (−1.07,0.92)	NMES	No data	0.24 (−0.11,0.59)
No data	No data	No data	No data	No data	Electroacupuncture	No data
0.21 (−0.20,0.62)	**−1.04 (−2.00,−0.08) ***	**−0.81 (−1.28,−0.34) ***	−0.32 (−1.25,0.61)	−0.24 (−0.59,0.11)	No data	Standard
**(D) Fugl–Meyer Assessment**						
rPMS	No data	No data	No data	No data	No data	No data
No data	Taping	No data	**1.75 (0.70,2.79) ***	**1.08 (0.31,1.85) ***	No data	0.97 (−0.06,2.00)
No data	No data	Steroid	No data	No data	No data	No data
No data	**−1.75 (−2.79,−0.70) ***	No data	Sling	−0.67 (−1.37,0.04)	No data	**−0.78 (−1.36,−0.20) ***
No data	**−1.08 (−1.85,−0.31) ***	No data	0.67 (−0.04,1.37)	NMES	No data	0.11 (−0.58,0.80)
No data	No data	No data	No data	No data	Electroacupuncture	No data
No data	−0.97 (−2.00,0.06)	No data	**0.78 (0.20,1.36) ***	−0.11 (−0.80,0.58)	No data	Standard
**(E) Shoulder PROM**						
rPMS	No data	No data	No data	No data	No data	No data
No data	Taping	No data	No data	No data	No data	No data
No data	No data	Steroid	**1.80 (1.15,2.46) ***	**1.91 (0.77,3.04) ***	No data	**2.48 (1.68,3.28) ***
No data	No data	**−1.80 (−2.46,−1.15) ***	Sling	0.10 (−0.82,1.03)	No data	**0.68 (0.21,1.15) ***
No data	No data	**−1.91 (−3.04,−0.77) ***	−0.10 (−1.03,0.82)	NMES	No data	−0.58 (−1.61,0.46)
No data	No data	No data	No data	No data	Electroacupuncture	No data
No data	No data	**−2.48 (−3.28,−1.68) ***	**−0.68 (−1.15,−0.21) ***	0.58 (−0.46,1.61)	No data	Standard

For efficacy, SMD > 0 favors the row-defining treatment, whereas SMD < 0 favors the column-defining treatment. Significant results are given in bold. Values in bold indicate statistically significant results. * indicates that the 95% CI does not cross 0. rPMS, repetitive peripheral magnetic stimulation; NMES, neuromuscular electrical stimulation; PROM, passive range of motion.

## Data Availability

The datasets used and/or analyzed in the current study are available from the corresponding authors upon reasonable request.

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
