# Peer review of "Comparative Effectiveness of Treatments for Shoulder Subluxation After Stroke: A Systematic Review and Network Meta-Analysis"

_jcm, 2025, doi:10.3390/jcm14196913_

Round 1
Reviewer 1 Report
Comments and Suggestions for Authors
Title:
- Appropriate.
Abstract:
- Not sure the sentence "This is the first synthesis to simultaneously compare diverse rehabilitation strategies for post-stroke shoulder subluxation" is needed in the conclusion. Also I don't think we can conclude anything about range of motion after reading results section within manuscript.
- Likely would be helpful to know in abstract that follow up was only 1-12 weeks, as any conclusions made would obviously be in the context of very short-term results.
Introduction:
- There's a lot of claims of being the first, but this review appears similar: Arya, K. N., Pandian, S., Vikas, & Puri, V. (2017). Rehabilitation methods for reducing shoulder subluxation in post-stroke hemiparesis: a systematic review*. Topics in Stroke Rehabilitation, 25(1), 68–81. https://doi.org/10.1080/10749357.2017.1383712. I think it's fine to point out novelties, but within abstract and introduction thus far there are many claims to being first which is rarely the case in 2025.
Materials and Methods:
- Appropriate.
Results:
- What does " not retrieved reports" mean?
Discussion:
- "The findings of Faghri et al. (1994) [29] and Koyuncu et al. (2010) [22], which reported that NMES effectively reduced shoulder subluxation and improved upper limb function, as well as those of Lavi et al. (2022) [24], which found that long-duration NMES decreased subluxation but had limited effects on pain relief, are consistent with the results of our study. Similarly, studies by Yim et al. (2024) [26] and Chatterjee et al. (2016) [27] demonstrated that shoulder taping improves functional outcomes, but has limited effects on subluxation reduction, which is consistent with our findings." I'm assuming none of these were the RCTs included in the meta-analysis, as similar findings between the meta-analysis and the studies included within it would be obvious. As opposed to other separate meta-analyses such as:
- Lee JH, Baker LL, Johnson RE, Tilson JK. Effectiveness of neuromuscular electrical stimulation for management of shoulder subluxation post-stroke: a systematic review with meta-analysis. Clin Rehabil. 2017 Nov;31(11):1431-1444. doi: 10.1177/0269215517700696. Epub 2017 Mar 27. PMID: 28343442.
- Systematic Review on Effectiveness of shoulder taping in Hemiplegia. Ravichandran, Hariharasudhan et al. Journal of Stroke and Cerebrovascular Diseases, Volume 28, Issue 6, 1463 - 147
- Wang et al.Efficacy of functional electrical stimulation for hemiplegic shoulder pain: A systematic review and meta-analysis
- Regarding limitations, the follow up of 1-12 weeks truly does limit what we can conclude from this study. We also did not have much regarding active or passive range of motion. The question remains too, what would clinically meaningful range of motion be in post-stroke patients?
Conclusion:
- Similar comment to abstract - can we really conclude anything regarding range of motion? You're also asking us to accept findings of comparisons between interventions with multiple studies versus interventions that had a single study each (i.e. local steroid injections, rPMS, and electroacupuncture.
References:
- Appropriate.
Tables and Figures:
- Appropriate.
Author Response
Dear. Editors and Reviewers,
Manuscript ID : jcm-3871253
Manuscript title: Comparative Effectiveness of Treatments for Shoulder Subluxation After
Stroke: A Systematic Review and Network Meta-Analysis
I greatly appreciate your review on my manuscript. We tried to make a comprehensive revision according to reviewer’s comments. The first manuscript revision is highlighted in red text.
< Recommendations of Reviewers >
- Reviewer 1 :
Abstract:
- Not sure the sentence "This is the first synthesis to simultaneously compare diverse rehabilitation strategies for post-stroke shoulder subluxation" is needed in the conclusion. Also I don't think we can conclude anything about range of motion after reading results section within manuscript.
: We sincerely thank the reviewer for these thoughtful comments. We agree with both points and have revised the abstract accordingly. Regarding novelty: We agree that reiterating the statement “this is the first synthesis” in the Conclusions was redundant, as the novelty is already highlighted in the Objectives. This sentence has therefore been removed to streamline the conclusion. Regarding range of motion (ROM): We agree that drawing firm conclusions about ROM was an overstatement, particularly since our network meta-analysis of shoulder PROM showed significant global inconsistency (χ² = 8.18, p = 0.004), requiring cautious interpretation. We have therefore removed references to ROM improvement from the Abstract Results and Conclusions. The revised abstract sections are shown below.
“(line 22-23) Objectives: This study conducted a systematic review and network meta-analysis to compare multiple therapeutic interventions for post-stroke shoulder subluxation,”
“(line 33-36) Results: Thirteen studies including 402 patients were analyzed. NMES was the most effective intervention for reducing subluxation distance (SUCRA: 84.9), while corticosteroid injections provided the greatest pain relief at rest (SUCRA: 73.6).”
“(line 38-39) Conclusions: Our network meta-analysis suggests that different interventions are optimal for specific aspects of post-stroke shoulder impairment.”
- Likely would be helpful to know in abstract that follow up was only 1-12 weeks, as any conclusions made would obviously be in the context of very short-term results.
: We thank the reviewer for this excellent suggestion. We agree that specifying the follow-up duration in the abstract is crucial for contextualizing our findings as short-term. To address this, we have added a sentence in the Methods section of the abstract clarifying that the included studies reported follow-up durations ranging from 1 to 12 weeks. In addition, to reinforce this limitation, we have slightly revised the Conclusions section to explicitly state that our results reflect short-term outcomes.
“(line 30-31) The follow-up duration in the included trials ranged from 1 to 12 weeks.”
“(line 42-46) These findings, based on short-term follow-up durations (1–12 weeks), provide an evidence-based ranking of interventions to support multimodal rehabilitation and inform clinical decision-making. The observed heterogeneity across studies underscores the need for standardized treatment protocols and rigorous long-term investigations.”
Introduction:
- There's a lot of claims of being the first, but this review appears similar: Arya, K. N., Pandian, S., Vikas, & Puri, V. (2017). Rehabilitation methods for reducing shoulder subluxation in post-stroke hemiparesis: a systematic review*. Topics in Stroke Rehabilitation, 25(1), 68–81. https://doi.org/10.1080/10749357.2017.1383712. I think it's fine to point out novelties, but within abstract and introduction thus far there are many claims to being first which is rarely the case in 2025.
: We thank the reviewer for this important observation. We agree that our initial manuscript overstated its novelty by repeatedly using the phrase “first.” As the reviewer correctly points out, prior systematic reviews (e.g., Arya et al., 2017) have already addressed rehabilitation methods for post-stroke shoulder subluxation. We have now cited this study in the Introduction.
The specific novelty of our work lies in the application of network meta-analysis (NMA), which allows for the integration of both direct and indirect evidence across multiple interventions and generates a comparative efficacy ranking (SUCRA). This methodological advancement differentiates our study from prior systematic reviews, which were limited to pairwise syntheses.
Accordingly, we have revised both the Abstract and Introduction to remove overstated claims of being the “first,” while clarifying the unique contribution of our study as the application of NMA to this clinical question.
“(line 76-84) While previous systematic reviews, such as that by Arya et al. (2018) [9], have summarized the available evidence, most randomized controlled trials (RCTs) have compared only a limited number of treatments, leaving uncertainty regarding their relative efficacy across the full spectrum of available options. To address this gap, our study applies a network meta-analysis (NMA) to simultaneously compare multiple therapeutic approaches. By integrating both direct and indirect evidence, this NMA provides an evidence-based hierarchy of treatment efficacy, offering insights that may guide clinical decision-making, optimize individualized rehabilitation strategies, and inform the development of future practice guidelines.”
Results:
- What does " not retrieved reports" mean?
: We thank the reviewer for requesting clarification on this term. The phrase “not retrieved reports (n = 1)” referred to a study that was identified during our search and appeared relevant, but for which the full-text article could not be obtained despite multiple attempts, including additional internet searches.
To avoid ambiguity and improve clarity for readers, we have revised both the Results section and the PRISMA flow diagram (Figure 1). The wording has been changed from “not retrieved reports” to “full-text could not be obtained.”
“(line 165) and full-text could not be obtained (n = 1).”
Discussion:
- "The findings of Faghri et al. (1994) [29] and Koyuncu et al. (2010) [22], which reported that NMES effectively reduced shoulder subluxation and improved upper limb function, as well as those of Lavi et al. (2022) [24], which found that long-duration NMES decreased subluxation but had limited effects on pain relief, are consistent with the results of our study. Similarly, studies by Yim et al. (2024) [26] and Chatterjee et al. (2016) [27] demonstrated that shoulder taping improves functional outcomes, but has limited effects on subluxation reduction, which is consistent with our findings." I'm assuming none of these were the RCTs included in the meta-analysis, as similar findings between the meta-analysis and the studies included within it would be obvious. As opposed to other separate meta-analyses such as:
- Lee JH, Baker LL, Johnson RE, Tilson JK. Effectiveness of neuromuscular electrical stimulation for management of shoulder subluxation post-stroke: a systematic review with meta-analysis. Clin Rehabil. 2017 Nov;31(11):1431-1444. doi: 10.1177/0269215517700696. Epub 2017 Mar 27. PMID: 28343442.
- Systematic Review on Effectiveness of shoulder taping in Hemiplegia. Ravichandran, Hariharasudhan et al. Journal of Stroke and Cerebrovascular Diseases, Volume 28, Issue 6, 1463 - 147
- Wang et al.Efficacy of functional electrical stimulation for hemiplegic shoulder pain: A systematic review and meta-analysis
: We sincerely thank the reviewer for this constructive feedback and for pointing us toward relevant literature. We apologize for the lack of clarity in our original phrasing, which indeed created confusion. The studies cited in that paragraph (Faghri et al., 1994; Koyuncu et al., 2010; Lavi et al., 2022; Yim et al., 2024; Chatterjee et al., 2016) were all included in our network meta-analysis. Our original intention was to illustrate how the results of individual RCTs supported the pooled estimates; however, we recognize that this wording appeared circular and was not helpful for interpretation.
We fully agree with the reviewer’s suggestion that a more meaningful comparison is to situate our findings within the context of other published systematic reviews and meta-analyses. Therefore, we have removed the original paragraph and replaced it with a revised discussion that directly compares our NMA results to previous meta-analyses, including those suggested by the reviewer. This revision, we believe, substantially improves the clarity and scholarly contribution of the Discussion.
“(line 311-329) Our findings are broadly consistent with, and extend, those of prior systematic re-views and meta-analyses. In line with Lee et al. (2017), our NMA identified NMES as the most effective intervention for reducing shoulder subluxation in acute and subacute stroke patients [5]. However, whereas our analysis did not demonstrate a significant benefit of NMES for pain relief, Wang et al. (2025) reported that functional electrical stimulation (FES), a subtype of NMES, was effective in alleviating hemiplegic shoulder pain [34]. This divergence may reflect differences in stimulation protocols, outcome measures, or patient populations, underscoring the need for further investigation into optimal NMES/FES dosing strategies. With respect to Kinesio taping, our NMA found it to be the most effective intervention for enhancing functional recovery and reducing pain during activity. These results partially align with the systematic review by Ravichandran et al. (2019), which concluded that taping reduces both pain and shoulder subluxation [35]. While both re-views support the role of taping in pain management, our analysis suggests that its primary benefits may lie in functional improvements and activity-related pain relief rather than structural correction of subluxation. By employing a NMA, our study synthesizes these findings within a single comparative framework and provides the first quantitative, hierarchical ranking of multiple interventions. This approach offers a more comprehensive guide for clinicians seeking to optimize rehabilitation strategies for post-stroke shoulder impairment.”
- Regarding limitations, the follow up of 1-12 weeks truly does limit what we can conclude from this study. We also did not have much regarding active or passive range of motion. The question remains too, what would clinically meaningful range of motion be in post-stroke patients?
: We thank the reviewer for these crucial and constructive comments, which highlight important limitations of both the current evidence base and our analysis. In response, we have substantially revised the Limitations paragraph in the Discussion section to emphasize that the short follow-up duration (1–12 weeks) restricts our conclusions to short-term effects and leaves the long-term sustainability of these interventions uncertain. We also clarified that our analysis was constrained by the scarcity of data on both active and passive ROM, rather than passive ROM alone. Finally, we added that the minimal clinically important difference (MCID) for ROM in post-stroke patients has not yet been established, which is essential for interpreting clinical relevance and should be a priority for future research.
“(line 408-419) More importantly, all included trials had short follow-up durations (1–12 weeks), restricting our conclusions to immediate or short-term effects and leaving the long-term sustainability of these interventions uncertain. As noted in the Methods, the analysis was constrained by limited data on ROM. The NMA could only be performed for shoulder ab-duction, as data for flexion and external rotation—movements critical for daily activities and pain mechanisms [44,45]—were rarely reported in the included trials. In addition, the minimal clinically important difference (MCID) for shoulder ROM in post-stroke patients has not been established, making it difficult to determine whether the statistically significant improvements observed translate into clinically meaningful benefits. Future studies should therefore employ standardized and comprehensive ROM assessments, establish clinically relevant thresholds for functional recovery, and investigate long-term outcomes.”
Conclusion:
- Similar comment to abstract - can we really conclude anything regarding range of motion? You're also asking us to accept findings of comparisons between interventions with multiple studies versus interventions that had a single study each (i.e. local steroid injections, rPMS, and electroacupuncture.
: We thank the reviewer for these critical and insightful comments and have revised the Conclusion section accordingly. We acknowledge that our previous conclusion overstated the evidence for ROM; given the statistically significant global inconsistency observed in the PROM network analysis (χ² = 8.18, p = 0.004), it is inappropriate to make a firm conclusion about the efficacy of any intervention for improving ROM, and we have therefore removed definitive statements on this outcome, consistent with revisions made to the Abstract. We also agree that the reliability of network meta-analysis estimates is reduced when some interventions are informed by only a single trial, and to clarify this limitation we have added a cautionary note in the Conclusion emphasizing that findings related to such interventions (e.g., local steroid injections, rPMS, and electroacupuncture) should be interpreted with caution.
“(line 422-430) This network meta-analysis provides an evidence-based hierarchy for managing post-stroke shoulder impairment, indicating that NMES is most effective for reducing subluxation, Kinesio taping for enhancing function and alleviating activity-related pain, and steroid injections for relieving pain at rest. However, these findings should be interpreted with caution, as the evidence for several interventions is derived from single studies, and no firm conclusions can be drawn regarding range of motion outcomes due to in-consistent data. Overall, our results support a tailored, multimodal rehabilitation approach that aligns therapeutic choices with specific patient goals while acknowledging the limitations of the current evidence base.”

Reviewer 2 Report
Comments and Suggestions for Authors
This is a well-designed systematic review and frequentist network meta-analysis of 13 RCTs (≈402 participants) evaluating interventions for post-stroke shoulder subluxation. The protocol was registered, PRISMA 2020 was followed, and the search was recently updated. Findings are clinically actionable: neuromuscular electrical stimulation (NMES) ranks best for reducing subluxation distance; corticosteroid injections provide the greatest relief for pain at rest and improve shoulder abduction PROM; and Kinesio taping shows advantages for functional recovery (e.g., FMA) and pain during activity. The authors are commended for a thorough risk-of-bias assessment, inconsistency checks (node-splitting and design-by-treatment), and clear presentation of SUCRA rankings across structural, pain, ROM, and functional domains. Overall, this manuscript fills with practical evidence gap for multidisciplinary stroke rehabilitation.
- You define this as a standard rehabilitation program (PT/OT) in Methods; please specify typical components (e.g., positioning, ROM, task practice) and dose to help readers interpret contrasts with active interventions.
- Table S3 indicates that intra-articular and subacromial routes were pooled as “steroid.” Consider a sensitivity/subgroup analysis (or at least a stronger limitation statement) to address potential heterogeneity from different targets/techniques.
- Egger’s test is significant (P=0.006). Please add a brief bias-impact discussion (and, if feasible, a trim-and-fill or selection-model sensitivity) in the main text.
- The global design-by-treatment test is significant (χ²=8.18, p=0.004). Please expand on clinical/statistical explanations and indicate how this affects confidence in SUCRA and pairwise estimates for PROM.
- You report three low-risk, eight “some concerns,” and two high-risk RCTs; please map these judgments to CINeMA downgrades per outcome in the main text (currently mostly in Supplement).
- Studies span acute to chronic phases; consider a stratified narrative (and, if power allows, subgroup) to explore timing-by-treatment interactions (e.g., early NMES effects).
- PROM analyses were limited to abduction—state this earlier (Methods/Abstract) and cross-reference as a limitation.
- Define AHI, AGT, ALT, SPADI, MBI, NPRS, etc., at first use in the main text; some appear first in the Supplement.
- Use one term—“Kinesio taping” vs “kinesiology taping”—consistently across text, figures, and tables.
- Where available, surface NMES (frequency, pulse width, session duration) and rPMS dosing details are in a concise main-text table for clinical reproducibility (now mostly in Table S3).
- Add one sentence explaining how to interpret SUCRA alongside clinically meaningful SMDs (especially where SUCRA is high, but CIs are wide).
- Ensure uniform figure labeling and p-value formatting (e.g., “P = 0.006”) across main and supplementary figures; check equal-contribution footnotes and author contact lines for duplication.
Author Response
Dear. Editors and Reviewers,
Manuscript ID : jcm-3871253
Manuscript title: Comparative Effectiveness of Treatments for Shoulder Subluxation After
Stroke: A Systematic Review and Network Meta-Analysis
I greatly appreciate your review on my manuscript. We tried to make a comprehensive revision according to reviewer’s comments. The first manuscript revision is highlighted in red text.
< Recommendations of Reviewers >
- Reviewer 2
This is a well-designed systematic review and frequentist network meta-analysis of 13 RCTs (≈402 participants) evaluating interventions for post-stroke shoulder subluxation. The protocol was registered, PRISMA 2020 was followed, and the search was recently updated. Findings are clinically actionable: neuromuscular electrical stimulation (NMES) ranks best for reducing subluxation distance; corticosteroid injections provide the greatest relief for pain at rest and improve shoulder abduction PROM; and Kinesio taping shows advantages for functional recovery (e.g., FMA) and pain during activity. The authors are commended for a thorough risk-of-bias assessment, inconsistency checks (node-splitting and design-by-treatment), and clear presentation of SUCRA rankings across structural, pain, ROM, and functional domains. Overall, this manuscript fills with practical evidence gap for multidisciplinary stroke rehabilitation.
- You define this as a standard rehabilitation program (PT/OT) in Methods; please specify typical components (e.g., positioning, ROM, task practice) and dose to help readers interpret contrasts with active interventions.
: We thank the reviewer for this insightful comment. We agree that a more detailed description of the "conventional treatment" is crucial for interpreting the comparative effectiveness of the active interventions. Accordingly, we have revised the Methods section to provide a clearer definition.
“(line 108-111) Conventional rehabilitation comprised passive and active range of motion exercises, stretching, positioning, strengthening, and task-oriented training, with some protocols al-so including posture management, balance or gait training, and complication prevention. Programs were typically administered 5 days per week for 30–180 minutes per day over 4–6 weeks.”
- Table S3 indicates that intra-articular and subacromial routes were pooled as “steroid.” Consider a sensitivity/subgroup analysis (or at least a stronger limitation statement) to address potential heterogeneity from different targets/techniques.
: We thank the reviewer for this critical observation. We agree that pooling different steroid injection techniques is a significant limitation. We have revised the limitations section in our discussion to address this point more thoroughly. The new text now explicitly states that pooling intra-articular and subacromial injections into a single "steroid" category may have introduced clinical heterogeneity due to differences in injection targets and techniques. We also clarify that because of the limited number of available trials, performing a subgroup or sensitivity analysis was not feasible. Consequently, we have added a statement urging that the findings regarding steroid efficacy be interpreted with caution. We believe this revision appropriately addresses the reviewer's concern.
“(line 397-402) In particular, intra-articular and subacromial steroid injections were pooled as a single category (“steroid”), which may have introduced clinical heterogeneity because of differences in injection targets and techniques. Given the limited number of available trials, subgroup or sensitivity analyses were not feasible, and our findings regarding steroid efficacy should therefore be interpreted with caution.”
- Egger’s test is significant (P=0.006). Please add a brief bias-impact discussion (and, if feasible, a trim-and-fill or selection-model sensitivity) in the main text.
: We thank the reviewer for highlighting the significant Egger’s test result and the need to address its implications. Following the reviewer’s suggestion, we conducted a trim-and-fill sensitivity analysis to further investigate the funnel plot asymmetry for the shoulder subluxation distance outcome.
The Egger’s test confirmed significant asymmetry (P = 0.006). However, the trim-and-fill analysis imputed zero studies, leaving the pooled effect estimate unchanged. This discrepancy suggests that the asymmetry is more likely attributable to substantial heterogeneity among the included studies rather than classical publication bias due to missing trials.
Although the point estimate was not altered by the trim-and-fill procedure, the significant Egger’s test still warrants a cautious interpretation of the findings for NMES. We have revised the Results (Section 3.2) and the Discussion (Section 4) to reflect this more nuanced analysis, acknowledging the presence of small-study effects and emphasizing that the observed asymmetry is likely driven by heterogeneity.
“(line 203-208) Funnel plots (Figure S2 (A–E)) and Egger’s test p-values indicated significant asymmetry for the shoulder subluxation distance (P = 0.006). However, trim-and-fill analysis did not impute any missing studies, and the pooled effect estimate remained unchanged, suggesting that the observed asymmetry is unlikely to be fully explained by publication bias alone and may instead reflect heterogeneity among the included trials.”
“(line 402-407) Specifically, for the shoulder subluxation outcome, a significant Egger's test suggested the presence of small-study effects. While a subsequent trim-and-fill analysis did not alter the pooled estimate—implying the asymmetry is likely driven by heterogeneity rather than missing studies—the potential for bias from small, influential trials remains. There-fore, the findings regarding NMES efficacy for subluxation must be interpreted with caution.”
- The global design-by-treatment test is significant (χ²=8.18, p=0.004). Please expand on clinical/statistical explanations and indicate how this affects confidence in SUCRA and pairwise estimates for PROM.
: We appreciate this important comment. The significant inconsistency for PROM shoulder abduction (p = 0.004) indicates conflict between direct and indirect evidence, violating the transitivity assumption of NMA. This likely reflects clinical heterogeneity (acute vs. chronic stroke populations, varied protocols, short follow-up) and sparse data. As a result, the pooled PROM estimates and SUCRA rankings are unreliable. We have revised the manuscript to clarify that PROM findings are exploratory only and should not be interpreted as definitive.
“(line 214-216), indicating violation of the transitivity assumption; thus, the pooled estimates and SU-CRA rankings for this outcome should be regarded as unreliable and exploratory only.”
“(line 265-268) The PROM results are presented for transparency but must be regarded as exploratory only, as significant global inconsistency (χ² = 8.18, p = 0.004) was detected in this network. Therefore, these findings should not be interpreted as definitive evidence of comparative efficacy.”
- You report three low-risk, eight “some concerns,” and two high-risk RCTs; please map these judgments to CINeMA downgrades per outcome in the main text (currently mostly in Supplement).
: We thank the reviewer for this suggestion. To improve transparency, we have revised the Results section to explicitly describe how the Risk of Bias (RoB2) assessments translated into CINeMA confidence ratings for each outcome. While the full tables remain in the Supplement (Tables S6–S7), the main text now summarizes the downgrading process and overall confidence levels per outcome.
“(line 291-297) According to the CINeMA assessment (Tables S6–S7), the certainty of evidence was generally moderate for shoulder subluxation distance and pain during activity, with downgrades mainly due to imprecision. Confidence for pain at rest and functional outcomes (FMA) was low, reflecting within-study bias (as captured by RoB2) and concerns about selective reporting. For PROM shoulder abduction, ratings were low to very low because of both significant network inconsistency and sparse data, limiting the reliability of these estimates.”
- Studies span acute to chronic phases; consider a stratified narrative (and, if power allows, subgroup) to explore timing-by-treatment interactions (e.g., early NMES effects).
: Thank you for this clinically important suggestion. We agree that the timing of intervention is a critical factor that could influence treatment efficacy.
A formal subgroup network meta-analysis based on stroke phase was not feasible due to the limited number of studies, which would result in sparse and statistically unreliable networks. Therefore, following your excellent alternative suggestion, we have added a stratified narrative to our Discussion section.
“(line 370-387) The included trials covered a wide range of stroke onset durations, from the acute phase (<1 month [26,30]) to the chronic phase (>12 months [27,33]). This heterogeneity likely contributed to differences in treatment responsiveness. In the acute and early sub-acute stages, NMES was consistently effective in preventing or reducing subluxation distance and improving motor activation [10,26,30]. These findings support the notion that early stimulation may help maintain glenohumeral alignment before irreversible capsular stretching and soft-tissue changes occur. In contrast, interventions tested predominantly in later phases showed a shift in therapeutic goals. Taping studies conducted in chronic populations [27] reported improvements in pain, active range of motion, and functional outcomes, but limited structural correction of subluxation. Similarly, steroid injection tri-als were conducted largely in chronic or late-subacute patients [33], with benefits focused on pain reduction and range of motion rather than alignment. These patterns suggest a timing-by-treatment interaction: NMES may be most beneficial when introduced early to prevent progressive subluxation, whereas taping and steroid injections appear more suit-ed to managing established pain and functional limitations in chronic stages. Although our dataset did not allow a statistically reliable subgroup network analysis, the stratified narrative underscores stroke phase as a plausible effect modifier and highlights the need for future RCTs designed to directly test timing-specific treatment efficacy.”
- PROM analyses were limited to abduction—state this earlier (Methods/Abstract) and cross-reference as a limitation.
: We thank the reviewer for this valuable suggestion. We agree that the restricted scope of our PROM analysis should be clarified earlier. We have therefore revised the Methods section (2.2) to state explicitly that the analysis was ultimately confined to shoulder abduction due to insufficient data for flexion and external rotation. In addition, the Discussion has been updated to cross-reference this point as a limitation. To maintain conciseness in the Abstract, and since no firm conclusions on PROM were presented there, we have not added this detail to the Abstract.
“(line 122-125) Another primary outcome was shoulder passive range of motion (PROM), including flex-ion, abduction, and external rotation; however, due to insufficient data for flexion and ex-ternal rotation, the final analysis was restricted to abduction.”
“(line 410-413) As noted in the Methods, the analysis was constrained by limited data on ROM. The NMA could only be performed for shoulder abduction, as data for flexion and external rotation—movements critical for daily activities and pain mechanisms [44,45]—were rarely reported in the included trials.
- Define AHI, AGT, ALT, SPADI, MBI, NPRS, etc., at first use in the main text; some appear first in the Supplement.
: We thank the reviewer for highlighting this important oversight. To improve clarity and ensure the manuscript is self-contained, we have revised the Methods section (2.2) where the study outcomes are first described. All outcome-related abbreviations (e.g., AHI, AGT, ALT, NPRS, SPADI, MBI) are now spelled out at first mention. This ensures that readers are familiar with all terminology before encountering it in later sections or in the Supplementary Materials.
“(line 114-122) The primary outcomes were shoulder subluxation distance, measured radiographically or ultrasonographically using indices such as the Acromio-Humeral Distance (AHD), Acromio-Humeral Interval (AHI), Acromion–Greater Tuberosity (AGT) distance, or Acromion–Lesser Tuberosity (ALT) distance, as well as by clinical palpation methods (e.g., fingerbreadth technique); pain, assessed with the Visual Analog Scale (VAS), Numerical Rating Scale (NRS), or Numerical Pain Rating Scale (NPRS); and functional recovery, primarily evaluated using the Fugl-Meyer Assessment (FMA), but also reported in some studies using additional scales such as the Shoulder Pain and Disability Index (SPADI) and the Modified Barthel Index (MBI).”
- Use one term—“Kinesio taping” vs “kinesiology taping”—consistently across text, figures, and tables.
: We thank the reviewer for identifying this inconsistency. To ensure clarity and consistency, we have standardized the terminology to “Kinesio taping” throughout the entire manuscript, including the main text and the supplementary materials (specifically Table S3).
- Where available, surface NMES (frequency, pulse width, session duration) and rPMS dosing details are in a concise main-text table for clinical reproducibility (now mostly in Table S3).
: We thank the reviewer for this excellent suggestion to improve clinical reproducibility. We agree that a concise summary of intervention parameters is highly valuable. Accordingly, we have extracted the available dosing details for NMES and rPMS from the included trials and compiled them into a new table (now Table 2) in the main text. In addition, we added a short paragraph in the Results section (3.1) to reference this table. This revision makes the intervention protocols more accessible to clinicians and researchers, while full details remain in Supplementary Table S3.
“(line 179-182) The intervention parameters for the NMES and rPMS trials are summarized in Table 1. Protocols varied considerably, particularly in session duration and frequency, with daily treatment times ranging from 20 minutes to several hours. Full details remain available in Table S3.”
- Add one sentence explaining how to interpret SUCRA alongside clinically meaningful SMDs (especially where SUCRA is high, but CIs are wide).
: We thank the reviewer for this important suggestion. We agree that SUCRA rankings must be interpreted in the context of effect size and precision. As requested, we have revised the Methods section (2.4) to clarify that SUCRA values should be considered alongside the standardized mean difference (SMD) and its 95% confidence interval (CI) to assess both the clinical significance and the certainty of the findings, especially when CIs are wide.
“(line 150-155) The ranking of treatment efficacy was determined using the surface under the cumulative ranking curve (SUCRA). However, SUCRA values should not be interpreted in isolation, as they provide only a relative hierarchy of interventions. They must be considered along-side the magnitude of the SMDs and the precision of its 95% CIs to determine the clinical significance and certainty of the findings, particularly when confidence intervals are wide [20].”
- Ensure uniform figure labeling and p-value formatting (e.g., “P = 0.006”) across main and supplementary figures; check equal-contribution footnotes and author contact lines for duplication.
: We thank the reviewer for this careful observation. We have systematically reviewed the manuscript and supplementary materials to ensure consistency. Specifically, all p-values have been reformatted to the style “P = 0.00X,” figure labeling has been standardized across the main text and Supplementary Figures, and the equal-contribution footnotes and correspondence lines have been consolidated to eliminate redundancy.
“(line 15-16) Correspondence: Sang-Chul Lee, bettertomo@yuhs.ac, Tel.: +82-2-2228-3711; Jae-Il Shin, shinji@yuhs.ac, Tel.: +82-2-2228-2073.”
“(line 18-19) Ç‚ These authors contributed equally to this work and are listed as co-first authors.
*These authors contributed equally to this work and are listed as co-corresponding authors

Round 2
Reviewer 1 Report
Comments and Suggestions for Authors
Appreciate the authors' edit and incorporate suggestions. No further edits noted.
Title:
- Appropriate.
Abstract:
- Appropriate.
Introduction:
- Appropriate.
Materials and Methods:
- Appropriate.
Results:
- Appropriate.
Discussion:
- Appropriate.
Conclusion:
- Appropriate.
References:
- Appropriate.
Tables and Figures:
- Appropriate.